# Lipid Metabolic Alterations in the ALS–FTD Spectrum of Disorders

**DOI:** 10.3390/biomedicines10051105

**Published:** 2022-05-10

**Authors:** Juan Miguel Godoy-Corchuelo, Luis C. Fernández-Beltrán, Zeinab Ali, María J. Gil-Moreno, Juan I. López-Carbonero, Antonio Guerrero-Sola, Angélica Larrad-Sainz, Jorge Matias-Guiu, Jordi A. Matias-Guiu, Thomas J. Cunningham, Silvia Corrochano

**Affiliations:** 1Neurological Disorders Group, Hospital Clínico San Carlos, IdISSC, 28040 Madrid, Spain; juanmiguel.godoy@salud.madrid.org (J.M.G.-C.); luiscarlos.fernandez@salud.madrid.org (L.C.F.-B.); mgilmoreno@salud.madrid.org (M.J.G.-M.); juaniglo@ucm.es (J.I.L.-C.); antonio.guerrero@salud.madrid.org (A.G.-S.); jorge.matiasguiu@salud.madrid.org (J.M.-G.); jordi.matias-guiu@salud.madrid.org (J.A.M.-G.); 2MRC Harwell Institute, Harwell Campus, Oxfordshire OX11 0RD, UK; z.ali@har.mrc.ac.uk (Z.A.); t.cunningham@prion.ucl.ac.uk (T.J.C.); 3Nutrition and Endocrinology Group, Hospital Clínico San Carlos, IdISSC, 28040 Madrid, Spain; angelica.larrad@salud.madrid.org; 4MRC Prion Unit at UCL, UCL Institute of Prion Diseases, London W1W 7FF, UK

**Keywords:** lipid metabolism, ALS, FTD, lipidomics, cholesterol

## Abstract

There is an increasing interest in the study of the relation between alterations in systemic lipid metabolism and neurodegenerative disorders, in particular in Amyotrophic Lateral Sclerosis (ALS) and Frontotemporal Dementia (FTD). In ALS these alterations are well described and evident not only with the progression of the disease but also years before diagnosis. Still, there are some discrepancies in findings relating to the causal nature of lipid metabolic alterations, partly due to the great clinical heterogeneity in ALS. ALS presentation is within a disorder spectrum with Frontotemporal Dementia (FTD), and many patients present mixed forms of ALS and FTD, thus increasing the variability. Lipid metabolic and other systemic metabolic alterations have not been well studied in FTD, or in ALS–FTD mixed forms, as has been in pure ALS. With the recent development in lipidomics and the integration with other -omics platforms, there is now emerging data that not only facilitates the identification of biomarkers but also enables understanding of the underlying pathological mechanisms. Here, we reviewed the recent literature to compile lipid metabolic alterations in ALS, FTD, and intermediate mixed forms, with a view to appraising key commonalities or differences within the spectrum.

## 1. Introduction

Lipids are essential components of life. The great complexity and number of lipid species is a major obstacle for understanding their extensive role in health and disease. The thousands of lipid species are currently classified into eight main families (fatty acyls, glycerolipids, glycerophospholipids, sphingolipids, sterol lipids, prenol lipids, saccharolipids, and polyketides) [1,2]. Fatty acyls are the core of certain complex lipid classes, including glycerolipids, glycerophospholipids and sphingolipids (Figure 1). Their functions and characteristics are mostly determined by their structure, which depends on the number of carbons of the chain (short, medium, long, or very-long fatty acids) and the number of double bonds (saturated, monounsaturated, and polyunsaturated or PUFA). In the brain, the majority of fatty acids are PUFAs that constitute the phospholipids of membranes. Interestingly, eicosanoids are the result of several oxidation processes of PUFAs, including arachidonic acid. These metabolites are involved in several inflammatory responses, directly linking the role of lipids and lipid peroxidation in the inflammatory processes observed in many disorders, including amyotrophic lateral sclerosis (ALS) and Alzheimer disease (AD). Similarly, the sphingolipids and ceramides, which are integral components of cell membranes and rafts, have important structural, signalling, and inflammation roles that depend, partly, on their fatty acyl composition. Fatty acids are stored in glycerolipids (mono-, di-, and triacylglycerols). Most of the triacylglycerols (TG) are composed of three fatty acyls with different carbons chains (16, 18, 20, etc.), thus explaining why many TG in blood have a total of around 52,54,56 carbons, with different quantities of double bonds. The major source of TG in the body is from the diet. Cholesterol levels in the body are tightly regulated, with 80% produced endogenously in the body, plus some coming from external sources. Cholesterol regulation can also be influenced by environmental interaction (diets, lifestyle, sport, etc.). Cholesterol can be found free or in an esterified form with fatty acyls (cholesterol esters, CE), with 18:2 CE and the 20:4 CE being the most abundant in human plasma, affecting the levels of PUFAs, such as arachidonate, in cells. Another lipid class is prenol. The most abundant and known prenol lipids are the benzoquinones CoQ9 and CoQ10, since they are essential for electron transfer in the mitochondria and ATP production. It is important to note that CoQ biosynthesis depends on the mevalonate pathway. HMG-CoA reductase, which is the rate-limiting enzyme in this pathway and in the biosynthesis of cholesterol, is inhibited by statins (cholesterol-lowering drugs), which might influence the reduction in CoQ levels in plasma [3] (Figure 1).

Thus, most lipid classes are interconnected. The study of the complex regulation of lipid metabolism, systemically and in neuronal tissues, has expanded since the introduction of lipidomics and other multiomics approaches in research that are clearly assisting in our understanding of these complex disorders.

In the CNS, lipids, and the fine regulation of lipid metabolism, are central for its correct development and function. Lipids constitute more than half of the dry weight of the brain, which is only second to adipose tissue in lipid content. In the adult brain, the most abundant type of lipid belongs to the glycerophospholipid family, which are the main structural components of cell membranes. These include phosphatidylserine (PS), phosphatidylethanolamine (PE), and phosphatidylcholine (PC), consisting of a head of acylglycerol with two fatty acids. The fatty acids vary among the phospholipids, but the most abundant in the adult brain are the polyunsaturated long fatty acids (PUFAs), such as the omega-3 docosahexaenoic acid (DHA; 22:6n−3) and the omega-6 arachidonic acid (AA; 20:4n−6) [4], e.g., phosphatidylserine (PS) is high in DHA [5]. Changes in the total quantity and the type of fatty acids and PUFAs impact on the properties of membranes, their fluidity, permeability, and signalling. The second most abundant lipid species in the adult brain belong to the cholesterol family. Up to 25% of the total cholesterol of the body is in the brain, and nearly 80% of that cholesterol can be found in myelin in the adult brain [6]. Cholesterol is delivered to the cells by lipoproteins, which are locally synthesised since they cannot pass the blood–brain barrier, especially during the developmental postnatal brain, with synthesis regulated by the myelination rate. Within the CNS, astrocytes and neurons exchange cholesterol thanks to important lipoproteins such as APOE and APOD. Excess cholesterol in the CNS is removed in the form of oxysterols (e.g., 24-hydroxycholesterol). The third most relevant type of lipid in the brain are the sphingolipids, which are mostly derived from the ceramides. Sphingomyelin is the major constituent of the cellular membranes and myelin sheaths. They can also form microdomains and lipid rafts that are crucial for cell communication and signalling.

Therefore, it is not surprising that alterations in the regulation of lipid metabolism in the CNS play major roles in neurological disorders. Lipid metabolism alterations are found in the majority of neurodegenerative disorders (NDDs) [7], including spinal muscular atrophy [8], spinocerebellar ataxia [9], in Huntington’s disease [10], Parkinson’s disease [11], and in Alzheimer’s disease [12,13]. At the same time, the lipidome of a healthy brain also varies with age, and even by brain region and cell type [14,15,16], with higher levels of lipids during the early decades of life and decreased later in aged brains. For example, the amount of total cholesterol is decreased in the healthy ageing brain, and most importantly in the frontal cortex [17]. Curiously, ages 50–55 is the time period when most changes seem to occur [18], which also coincides with whole-body metabolic changes in fat mass, body composition, and energy expenditure [19]. These changes are notable in that they coincide with the average age of onset of neurodegenerative disorders such as amyotrophic lateral sclerosis (ALS) and frontotemporal dementia (FTD).

ALS is a fatal, rapid, progressive neurodegenerative disorder characterised by loss of upper and lower motor neurons of the frontal cortex and spinal cord, leading to a loss of capacity to move, talk (dysarthria), swallow (dysphagia), and ultimately breath (due to weakness of the respiratory muscles), requiring tube feeding and mechanical ventilation at late stages [20,21]. Death typically occurs within 3–5 years from diagnosis and there is no cure [22,23]. Some drugs are clinically available, including riluzole and edaravone, but have shown only a modest effect on survival [24], and no effective disease modifying therapy is available to date. We know of > 30 genes that when mutated can cause dominant or recessive heritable ALS, although this only accounts for ~10% of patients—the cause for the remaining 90% of largely sporadic cases is currently unknown. Frontotemporal dementia is a group of dementias characterised by degeneration of neurons in the frontal and/or anterior temporal lobes of the brain, mainly leading to alterations in behaviour and/or language [25]. In FTD, familial cases are more prevalent (around 30–40%) [26] than in ALS, although most cases are also sporadic. There is a strong overlap in the genes that cause ALS and some forms of FTD; for example, the hexanucleotide GGGGCC (G4C2) repeat expansion in the first intron of the *C9ORF72* gene [27,28] is the most common genetic cause of both disorders [28,29]. Conversely, ALS patients with mutations in SOD1 are not expected to develop FTD, whilst FTD tauopathy is not related to motor symptoms. These diseases frequently co-occur in individuals, but the severity of each disorder in individuals varies. For example, some patients present with only motor alterations (pure ALS forms with no apparent dementia associated). At the other end of the spectrum, there are patients with only behavioural and language alterations (pure FTD forms with no apparent motor alterations). In addition, intermediate clinical manifestations occur, with both motor and cognitive-behavioural alterations at different degrees of severity (around 15% of FTD cases show motor alterations [30] and, similarly, more than 50% of ALS cases develop signs of cognitive impairments, with 15% of ALS cases diagnosed with FTD at onset [31,32]). Strikingly, ALS–FTD manifestation can vary within family pedigrees, suggesting non-genetic factors play a part.

Alterations in weight and lipid metabolism are evident in ALS, and it is becoming clear that such alterations are not only a consequence of muscle atrophy and the degenerative process, but are influential in the degree of risk for developing ALS, although underpinning the pathological mechanisms at play requires further research. In relation to FTD, there are now some studies showing metabolic alterations in patients, but the number of studies on lipid metabolism and FTD are much more limited than in ALS. Here, we search for the most relevant and recently published research on the lipid metabolic alterations (both systemically and in the CNS) in the complex ALS–FTD spectrum. We present the most updated findings on the weight, adiposity, lipid analysis (including lipidomics), and genomics regarding lipid metabolism alterations in ALS and FTD (Figure 2).

## 2. Amyotrophic Lateral Sclerosis and Lipid Metabolism

Alterations in the correct regulation of lipid metabolism in ALS are becoming evident [33]. Those alterations can be observed as weight and fat depot changes in patients. Finally, those alterations can be evidenced using more specialised techniques, such as transcriptomics and lipidomics, in blood and in neuronal tissue.

### 2.1. Weight

Epidemiological studies have revealed that ALS courses with weight loss, even before the onset of motor symptoms in many cases. It has also been shown that BMI is an independent prognostic factor for survival, and the rate of weight loss predicts the progression and severity of the disease [34,35], where more rapid weight loss has a worsening prognosis and survival [34,36,37]. Conversely, patients with no weight loss or weight stabilisation, or patients that were overweight at onset, have a slower disease progression and better prognosis.

Hypermetabolism is present in at least 40% of patients [38,39]. There are increasing studies pointing towards a causal dysregulation of hypothalamic networks that control energy metabolism in the body [40,41], although it warrants further research to draw a conclusive picture. At the same time, ALS patients, especially those with bulbar forms, can progress rapidly due to difficulties with swallowing, chewing, and digestion (dysphagia), resulting in progressive reduced food intake and associated malnutrition that greatly contributes to weight loss at later stages. As expected, there is a negative association between nutritional status and the progression of disease in ALS [42,43].

Whilst the associations of weight loss and ALS prognosis and survival are clear during the course of disease, and can be easily observed and studied, pre-diagnosis association with weight or BMI has been harder to ascertain, relying in part on historical records that may be limited and questionnaires and interviews of the patients. Such studies into pre-symptomatic associations are relatively recent, and many within the control populations are still alive; thus, it is not known whether they could develop ALS later in their lives. Hence, there are some inconsistent conclusions from population studies. Nevertheless, most studies point towards the protective role of a higher BMI before disease onset. Using very large population prospective studies, such as the European Prospective Investigation into Cancer and Nutrition (EPIC) [44] and the Pooling Project of Prospective Studies of Diet and Cancer (DCPP), ALS-research collaboration has been established. In the pan-European prospective cohort (EPIC), out of approximately 500,000 people in the study, 222 developed and died of ALS, and they found a suggestive association between higher BMI at pre-diagnosis and lower risk of ALS [44]. In another collaborative analysis of ten prospective cohorts with 568,070 participants from the general populations of Europe, the US, and Australia, there were 428 women and 204 men who had ALS listed as their cause of death for the analysis, and the pre-diagnostic BMI and waist-to-hip ratio (WHR) were inversely associated with ALS mortality. For a 5 kg/m^2^ increased BMI, the disease progression rate was 15% lower [45]. From the US National Registry of Veterans, there were 467 patients with ALS and 975 matched controls selected for a study of the impact of pre-diagnosed BMI in ALS, and although there was a positive relation between lower BMI in middle age and higher ALS risk, they found no association between pre-diagnostic BMI and survival [46]. Recently, it has been shown in a longitudinal population-based study (a 30–50 years follow-up study of Norwegian tuberculosis screening program), with nearly 3000 ALS cases detected, that high pre-diagnostic BMI was associated with lower risk of ALS [47]. Thus, it supports previous observations that a higher BMI is a protective factor in ALS (Figure 3).

### 2.2. Adiposity

In terms of adiposity, it has long been accepted that weight loss and lower BMI is strongly correlated with the loss of fat mass (FM). In ALS, there are reports that FM is reduced both in ALS patients and in animal models of ALS [48], and that the rate of this reduction is associated with faster disease progression [49]. This led to studies in mouse models of ALS that found increased lipolysis (mainly in adipose tissue) supported by the finding of elevated plasma free fatty acid (FFA) in patients with ALS. Unfortunately, there are a very few studies that have measured the distribution, amount, and functionality of adipose tissue in ALS patients. Many of those studies in ALS patients are indirect, using body adiposity index, which provides a poor estimate of the fat mass in ALS. Ioannides et al. studied fat mass in ALS by air displacement plethysmography (FM-ADP), describing how loss of BMI is not associated with a loss of FM, but rather with muscle atrophy and fat redistribution [50]. Similarly, another study found that total body fat in ALS patients was not different from healthy controls, but rather the ratio of abdominal subcutaneous versus visceral fat distribution, measured by MRI, was lower. In this study, ALS patients showed a higher proportion of visceral fat than subcutaneous fat [51], resembling the pattern observed in people with higher risk of metabolic disorders. Another observation in some ALS patients is the higher presence of accumulated fat in the liver [52]. All these studies suggest that the total fat mass of the body can be redistributed and differentially accumulated in organs in ALS, contributing to the systemic metabolic alterations observed in these patients (Figure 3).

### 2.3. Lipid Analysis (Classical, Lipidomic and Other -Omics Analysis)

A major indication of altered metabolism in ALS is the altered levels of circulating lipids; studying such changes may not only provide clues to better understand disease pathomechanisms but also identify biomarkers useful in a clinical setting.

There are many lipid species and intermediates that are not detected by conventional analytical tools. The development of quantitative lipidomics clearly aids our understanding of their characterization and functions. Until the recent development of lipidomics, the reported analysis of altered lipid species in disease situations have been limited to the easily detected and known major lipids, such as total cholesterol (TC), low-density lipoprotein (LDL), high-density lipoprotein (HDL), or triglycerides (TG), and in the great majority of cases those were analysed in the accessible plasma samples of patients.

In recent years, more in-depth lipidomic studies have allowed for identification of a more complex profile from patient blood, cerebrospinal fluid (CSF), and even neuronal tissue samples.

We gather the main findings regarding the major types of lipids identified in the blood, CSF, and neuronal tissues in ALS, analysed by classical biochemistry and most recent lipidomic techniques.

#### 2.3.1. Sterol Lipids

In blood samples, higher levels of LDL cholesterol (and TG) in a cohort of 369 ALS patients and 286 healthy controls were associated with increased survival in ALS (by more than 12 months) [52]. These results have been replicated with a recent cohort of 99 ALS patients, again showing that higher cholesterol, LDL/HDL, and apolipoprotein B levels in blood are associated with a lower risk of death after amyotrophic lateral sclerosis diagnosis [53]. Dyslipidaemia has also been reported in another case-control retrospective study of 650 ALS patients versus 365 controls, with higher LDL/HDL levels associated with disease duration, but not with disease progression [54]. However, a meta-analysis using blood lipid data from over 400 ALS patients in China [55] could not identify clear changes in cholesterol, TG, LDL, or LDL/HDL levels. Consistent with those results, a recent longitudinal case-control lipidomic analysis of blood samples of ALS patients versus controls identified higher levels of free cholesterol, and no changes in the total amount of cholesterol ester (CE) species, except for increased levels of CE (24:2) and CE (24:5) [56]. Finally, higher levels of HDL cholesterol have been associated with poor prognosis in ALS patients, especially on those with hypermetabolism [57]. This study has the limitation of a single-centre retrospective study with only 78 ALS patients. A large matched case-control nested study of prospective US cohort, the pre-diagnosed blood sample analysis of 275 individuals that developed ALS later, showed that higher levels of HDL cholesterol were statistically associated with higher risk of ALS [58], thus suggesting HDL-C as a potential biomarker assisting in the diagnosis of ALS. Further studies in different cohorts and populations are needed to confirm these results on HDL and LDL values and the origin of those alterations in ALS patients.

The study of CSF is considered a much closer representation of altered biological pathways in the degenerating spinal cord than the blood. In a lipidome study of the CSF of ALS patients versus controls, cholesterol levels were increased but the level of some of its precursors were decreased [59].

The study of specific lipid metabolic alterations more directly, in affected neuronal tissue, has also been very informative. Accumulation of cholesterol esters (CE) have been found in the spinal cord of ALS patients, as much as 22-fold for some CE species [60,61]. Interestingly, HMG-CoA reductase, the rate-limiting enzyme in cholesterol synthesis, was reduced in the grey matter spinal cord of ALS patients [61]. By using mouse and rat models carrying particular ALS causative mutations, some groups have been able to provide some confirmation of those findings as well as proposing novel altered lipid metabolic pathways involved in pathological degeneration in the spinal cord. In the spinal cord of a FUS transgenic model [62], a lipidomic analysis found accumulation of cholesterol esters, similar to previous findings in the spinal cord of a rat transgenic SOD1^G93A^ model [63]. The combination of several transcriptomic datasets at early and late disease stages in the spinal cord of SOD1^G93A^ transgenic mice have revealed that the endogenous cholesterol biosynthesis pathway is mainly downregulated while transport of cholesterol outside the cells is upregulated [64]. This downregulation of cholesterol production and upregulation of cholesterol export could be a compensatory mechanism of the higher cholesterol levels found in degenerative spinal cords.

Oxysterols are derived from the oxidation of cholesterol, and, unlike cholesterol, these metabolites can pass the blood–brain-barrier. Oxysterols deserve especial mention since they have recently been closely related to the pathology of ALS, specially 24S-hydroxycholesterol (24-OHC), 25-hydroxycholesterol (25-OHC), and 27-hydroxycholesterol (27-OHC). These metabolites are formed as a result of enzymatic reactions or by auto-oxidation. The 25-OHC levels are increased in blood and CSF from ALS patients [65], and the gene expression levels of the enzyme cholesterol 25-hydroxylase (CH25H) that converts cholesterol in 25-OH is upregulated from very early pre-symptomatic disease stages (60 days) in the spinal cord of SOD1^G93A^ mice [64]. The 24-OHC levels are found increased in post-mortem tissue and decreased in the CSF of ALS patients [66]. Finally, the 27-OHC levels are found significantly decreased in plasma of ALS patients [67], and the gene that encodes the enzyme that converts cholesterol into 27-OHC, *CPY27A1*, has been identified as a susceptibility gene for ALS [68].

Due to the alterations in blood cholesterol levels observed in ALS patients, several groups have evaluated the treatment efficacy of cholesterol-lowering statins in the risk of ALS. There are some controversies in these results, with mainly no effect, or a negative effect, on the progression of the disease [69,70]. Similarly, in mice, the use of statins to lower cholesterol levels accelerates disease progression and decreases survival in the SOD1^G93A^ transgenic models [71]. These results may be surprising since statins have been described as one of the most potent axonal regeneration-promoting compounds [72], and also are being considered for other neurodegenerative disorders since they also have anti-inflammatory and anti-oxidant properties, as well as attenuate the amyloid and protein aggregates [73]. The unexpected negative effects of statins in ALS patients may be due to the fact that increased blood cholesterol could be a beneficial compensatory mechanism. In this regard, a higher LDL/HDL ratio in ALS patients correlates with a better prognosis of the disease [52]. A significant percentage of patients develop hypermetabolism and this increase in cholesterol could supply the increased energy demands. With these data, treatments that seek to reverse the increase in blood cholesterol, such as statins, would not be the most beneficial, as studies to date have shown [69,70]. Additionally, the cholesterol biosynthesis pathway in the spinal cord of SOD1^G93A^ mice is transcriptionally downregulated [64], which might explain why the cholesterol-lowering statins have a deleterious effect on survival in these mice.

#### 2.3.2. Glycerolipids, Fatty Acyls, and Eicosanoids

Together with sterols, fatty acid and acylglycerol dysregulation have contributed greatly to the dyslipidaemia observed in ALS patients. When using non-omics techniques for the analysis of plasma samples of ALS patients, an increase in total fatty acids is observed, without changes in the TG levels [53,74,75]. In a meta-analysis using blood lipid data from over 400 ALS patients in China [55], there was a tendency of a better prognosis in patients with higher TG levels in their blood (a median prolonged life expectancy of 5.8 months for patients with serum TG levels above the median of 127.5 mg/dL). It is important to consider the sex in these studies, as evidenced by the differential levels of lipids in serum identified in women with ALS, in particular higher levels of cholesterol and TGs [76].

In contrast, the use of lipidomics in plasma samples of ALS patients reported increased levels of total TGs and diacylglycerides (DGs) [56,77], and decreased monoacylglycerides (MG) at baseline and at later stages of disease, in particular lower levels of MG (16:0, 18:0) species [56,78]. An important and consistent finding in the blood lipidome by different groups is the increased levels of very-long-chain fatty acids (VLCFA) in ALS patients [56,77,78]. There are some lipid profiles that seem to correlate with disease progression in ALS patients; in particular, faster progressive types showed decreased TG among other lipids involved in sphingolipid and glycerophospholipid metabolism in blood [79], which represent potential biomarkers that could help with the stratification of patients. A cross-sectional study of an untargeted lipidomic profile in plasma from 20 ALS patients compared to 20 healthy control individuals [77] identified a profile of altered levels of specific lipids, namely, two monounsaturated FAs, (C24:1n−9) and (C14:1), and the triglyceride TG (51:4), that could potentially be used to discriminate ALS patients from healthy controls. Onset site also seems to have a differential pattern, with bulbar-onset ALS patients showing higher levels of triacylglycerol TG (58:11) [79]. Similarly, there are blood lipid profiles that help to differentiate ALS from primary lateral sclerosis (PLS); specifically, changes in TG (50:3/16:1), as observed in a longitudinal case-control study [56].

In a lipidomic analysis of CSF from ALS patients, a decrease in MG (18:0) level was also reported, similarly to those identified in the blood of ALS patients. Contrary to the blood lipidome, in the spinal cord of ALS patients, the levels of total TG are mostly decreased [61,80], except for a few TG species that increased three-fold, such as TG with the saturated fatty acid palmitate (C16:0) and the TG with the monounsaturated fatty acid (MUFA) oleic acid (C18:1n9). Compared to the brain, spinal cords from ALS patients showed significant decreases in the content of PUFA, especially the DHA (C22:6n−3), while increases were detected in the frontal cortex [81].

Most recently, -omics combination analyses (transcriptomic and metabolomics) have been applied to several lines of motor neurons derived from patient hiPSCs lines (human-induced pluripotent stem cells). Interestingly, a comparative analysis of the SOD1-A4V, C9ORF72, TDP43-Q343R, and sporadic ALS-induced motor neuron lines revealed glycerophospholipid metabolism alterations, and most specifically, the activation of the arachidonic acid (AA) pathway, as the common metabolic signature [82]. Arachidonic acid is a PUFA of the omega-6 class of lipid. The activation of the AA pathway has been previously reported in the spinal cord of SOD1 mice at early disease stages [64], whilst phospholipase A2, which catalyses the conversion of AA, has also been found upregulated in human ALS patients [83], and its inhibition seems to be beneficial in SOD1^G93A^ mice [84]. Similarly, by targeting lipoxygenase 5-LOX—which inhibits the AA pathway in several motor neuron derived cell lines, in a drosophila C9ORF72 model (that overexpresses 30× G4C2 repeats), and in the mouse SOD1^G93A^ model—with caffeic acid, the authors could delay motor neuron loss [82].

#### 2.3.3. Sphingolipids and Ceramides

The metabolism of sphingolipids is consistently reported altered by pathway enrichment analysis of the dysregulated lipids found from ALS patients’ blood samples [79]. Specific alterations in the blood lipidomes of ALS patients included sphingomyelin SM (36:2) [77], and decreased levels of SM (22:1) and SM (20:1) [56]. In addition, there is an early and significant reduction in the total content of sphingomyelin (SM) in ALS that progressively worsens. This initial change in SM was followed by significant increases in other sphingolipid classes, such as lactosylceramides and globosides [56].

In the spinal cord of ALS patients, abnormalities in sphingolipid metabolism have also been reported, namely, increased levels of sphingomyelin and ceramides [60] (Ceramides C16:0, 24:0 and sphingomyelin SM (16:0), and a significant accumulation of several glycosphingolipids as cerebrosides, GalCer, GlcCer, and LacCer [85]. Some of these complex sphingolipids were also elevated in the CSF of patients [86].

Sphingolipid metabolism is also deregulated in the spinal cord of SOD1 mice. An RNA-seq study of SOD1^G86R^ mice spinal cord found overexpression of genes involved in the recycling of sphingolipids in the lysosome [87]. In addition, genes involved in the catabolism of glycosphingolipids were found to be upregulated, with a consequent increase in ceramide synthesis [64], in a meta-analysis of transcriptomic studies of the spinal cord of SOD1 mice spinal cords, which is consistent with the accumulation of ceramides observed in the spinal cord of ALS patients [60].

The combination of transcriptomic with metabolomic and lipidomics provides a higher level of information and complexity. Parallel lipidomics and transcriptomics studies have uncovered accumulation of cholesterol esters, sphingomyelin, and ceramides in ALS patient spinal cords [60]. Similarly, the combination of lipidomics with transcriptomics helped to pin down particular altered lipid metabolic pathways in the spinal cord of a FUS transgenic model [62]. Glycerophospholipid and sphingolipid metabolism, and cholesterol ester accumulations, were most affected, in agreement with other studies. These authors then targeted histone deacetylase (HDAC), and that mitigated the glycerophospholipids alterations and reduced the accumulation of cholesterol esters in the FUS transgenic mice [62].

#### 2.3.4. Glycerophospholipids

Lipidomic analysis of blood from ALS patients found alterations in the main classes of phospholipids with a significant decrease in phosphatidylcholine (PC), ether phosphatidylcholine (PCe), phosphatidylserine (PS), and plasmalogen phosphatidylethanolamine (Pep). Among all glycerophospholipid classes, PC and PS seem to be particularly and progressively decreased in ALS plasma compared to the controls [56,78]. In contrast, in the CSF there was a generalised increase in the phospholipid levels, especially of several PC species [79,88]. The main changes found in ALS patient CSF were dysregulated biosynthesis of glycerophospholipids and sphingolipids, with increased levels of phosphatidylcholine PC (36: 4) and other ceramides and glycosylceramides [88]. In a later CSF lipidome study, the main finding was a decrease in the monoacylglycerol (18:0) levels, and, consistent with the previous study, increased levels of PC (36:4) [79], which seems the most relevant lipid species to discriminate against ALS cases. There is a lipidomics study in the brain of sALS patients in comparison to FTLD-TDP patients in which the PC levels are significantly decreased compared to the controls [89].

A summary of the main and more consistent findings regarding lipid analysis in ALS can be found in Table 1.

### 2.4. Genetics

Several genome-wide association studies (GWAS) and exome and whole genome sequencing studies have been conducted in ALS, providing a large amount of information that can be used to identify lipid metabolic genes associated with ALS risk. We can use an informatics tool named mendelian randomization to infer causal relation between a risk factor and disease by using summary statistical results from different genome-wide association studies (GWAS). Using mendelian randomization studies to evaluate the causal relation of particular blood lipid levels (total cholesterol, LDL, HDL and TG) and the risk of sporadic ALS in the European and East Asian populations, only higher LDL levels are positively causally associated with an increased risk of ALS in several studies [96,97,98]. There are some other studies reporting an increasing association of risk-susceptible genes in sporadic ALS, such as the *CYP27A1* gene in the cholesterol pathway [68]. The importance of this finding is that it suggests that amongst the dyslipidaemia found in ALS patients, high LDL levels could have a genetic component, suggesting causation rather than correlation. This goes in agreement with the finding that a one-unit increase in LDL cholesterol was associated with a higher incidence of ALS in a population-based study in Sweden [99].

In relation to body weight and weight loss in ALS, single nucleotide polymorphisms (SNPs) in the *ACSL5* gene have been associated with lower fat-free mass in ALS patients [100]. This gene encodes a long-chain fatty acid coenzyme A ligase, important for fatty acid degradation that has previously been associated with weight loss in the general population [101]. Similarly, using mendelian randomization analysis, it has been shown that genetically determined higher adiposity and BMI suggestively decreases the risk of developing ALS [102], although these results have not been supported so far by others [98].

Recently, an epigenome wide association study (EWAS) using blood samples from 6763 ALS patients and 2943 controls, identified around 43 methylated loci that were associated to genes that participate in the cholesterol biosynthesis process, metabolism and inflation [103]. Since DNA methylation could be the result of gene-environmental complex interactions and disease progression, those findings support the concept that the metabolic changes observed in ALS patients could be the combination of previous exposures to environmental risk factors and the degenerative disease.

Altogether, it seems that there may still be some hidden complex genetic factors that could influence lipid metabolism and infer altered ALS risk. In this regard, LDL levels of cholesterol and body composition seem to be more genetically determined in ALS patients rather than only a mere consequence of the degenerative process.

## 3. FTD, Mixed Forms (ALS/FTD) and Lipid Metabolism

There are three clinical subtypes of FTD: the behavioural variant (bvFTD), the non-fluent/non-grammatical progressive primary aphasias (PPA), and the semantic dementia. Not all the clinical FTD forms are part of the dementia-motor neuron disease continuum. The bvFTD is the one that is more frequently found in that continuum. Pathologically, there are TDP43, C9ORF72 and FUS protein aggregate hallmarks that can be found in the bvFTD and FTD-ALS mixed forms. On the contrary, FTD with TAU pathology are not related to motor neuron diseases and are not considered part of the spectrum ALS–FTD [104]. Some studies concerning metabolism in FTD made no distinction in the clinical forms selected in their analysis, or the predominant pathological protein (TDP43 or TAU). Thus, the interpretation of lipid metabolism results in relation to FTD/ALS spectrum might be limited.

### 3.1. Weight

Contrary to observations in ALS, patients suffering FTD, in particular the behavioural variant (bvFTD), showed increased weight (predominantly reported as BMI) [105]. Comparing AD with the two forms of FTD (bvFTD and semantic variant FTD), and with a group of healthy people, the bvFTD patients had higher BMI than any of the other groups in the study [106]. Interestingly, there are significant differences in BMI across the ALS–FTD spectrum with higher BMI (overweight or obese) in bvFTD patients at one end of the spectrum; low BMI (underweight or normal weight) at the other end of the spectrum in patients with pure ALS forms; and a tendency to overweight in the mixed forms including the ALS-plus and the mixed ALS–FTD group [107] (Figure 3). The higher weight observed in the bvFTD patients is partly attributed to the associated hyperphagic phenotype, even though some might also present hypermetabolism at rest [90]. Supporting this idea, atrophied neuronal networks have been found to be associated with bvFTD patients, which could be directly associated with such hyperphagic behaviour [105].

Unfortunately, FTD has not been as extensively studied as ALS, and so there are less epidemiological studies that could reveal the role of lifelong weight and other lipid metabolic factors associated with disease risk. In one population-based, longitudinal nested case–control study, with 90 patients with FTD, 654 patients with AD, and 116 individuals in the control group, weight data prior to the debut of the disease was recorded with self-assessment questionnaires. There was a significant association of obesity (especially in the middle age) and the risk of FTD [108]. However, the data were related to the prodromal phase; this association may be explained by the data that were related, where there were already changes in food preference and eating habits. Another case–control study, with around 100 FTD patients and 200 healthy age–sex matching controls, found no cardiovascular risk factors associated with FTD, except for diabetes mellitus [109].

Thus, the association of higher BMI and FTD is clear, but it seems as one of the consequences for the hyperphagic and eating habits observed in these patients, whereas the role of high BMI as a risk factor for FTD is not that clear and needs further research.

### 3.2. Adiposity

bvFTD patients have a higher weight and BMI and that promotes bigger fat depots, measured by indirect impedance [105], in particular visceral fat accumulation. Thus, FTD patients seem to have increased adiposity. It would be interesting to conduct further analysis on different types of bvFTD, stratifying patients by their genetic cause, where known, or by pathological protein hallmarks (e.g., TDP-43, TAU), and certainly by clinical presentation, and whether patients suffer from pure bvFTD or mixed forms with additional motor alterations (ALS–FTD mixed forms).

### 3.3. Lipid Analysis (Classical, Lipidomic, and Other -Omics Analysis)

The analysis of general lipids by classical clinical biochemistry in blood and plasma has shown that patients with bvFTD present higher levels of TGs, which seems to be correlated with higher BMI [106]. The levels of TGs in blood are not only a consequence of the body’s lipid metabolic regulation but also a reflection of the person’s diet, since nearly two thirds of TGs are diet derived. Since TGs can pass the blood–brain barrier, it is important to consider their levels in blood and reflect the impact that those might have on the brain. Contrary to ALS, LDL cholesterol levels are not altered in the blood of FTD patients. On the other hand, there is some evidence that HDL cholesterol levels are decreased in bvFTD patients [90,91]. Unfortunately, another study could not corroborate these findings and reported higher levels of all cholesterol species analysed in blood [92]. There is therefore controversy between the results obtained in the different studies, indicating that there may be other factors contributing at the population level.

There are very few comprehensive unbiased lipidomics analyses conducted in FTD patients to date. One of the earliest unbiased lipidomic analyses was conducted on blood samples of 16 bvFTD patients compared to 14 AD and 22 healthy controls [93]. This lipidomic analysis corroborated not only the higher TG levels previously described in blood with classical methods but also determined a more descriptive paradigm of the lipid species in bvFTD patients. Those include mainly five lipid classes: for the TGs (16:0) then DGs (18:1/22:0), phosphatidylcholine PC (32:0), phosphatidylserine PS (41:5), and sphingomyelin SM (36:4). Those changes were suggested by the authors as potential biomarkers for bvFTD [93]. A follow-up lipidomic analysis, from blood samples of nearly 40 patients of bvFTD and 22 healthy controls, corroborated the higher levels of total TG in bvFTD patients, and identified differences in other lipid species; in particular, lower levels of cardiolipin and higher levels of inflammatory lipids [94]. Regarding fatty acids, which are the main components of the TGs, they found higher levels of unsaturated fatty acids, both in blood samples and post-mortem brains of FTD patients versus the controls. Since these fatty acids are more prone to lipid peroxidation, the authors then analysed lipid aldehyde by-products, detecting higher levels of proteins conjugated with acrolein (a lipid aldehyde) in blood and brain samples of bvFTD patients [94].

Another targeted lipidomic study, this time directly in post-mortem brain samples from FTD patients, was investigating very-long fatty acid lipids (VLCFA lipids, contain > 26 carbon atoms in the fatty acid chain) in FTD, since higher levels of VLCFA lipids are toxic and can cause early dementia [110,111]. This study found that FTD patients do have higher VLCFA lipid levels in the brain, plus the two enzymes that metabolise them (ELOVL4 and the transporter ABCD1), suggesting that such alterations in lipid metabolic regulation could be a central pathogenic mechanism of this disorder [95].

An interesting study aimed to evaluate whether there is a pattern in relation to alterations in lipid metabolic pathways among the spectrum of TDP43 proteinopathies and their corresponding clinical phenotypes: ALS patients, sporadic FTLD, c9FTLD patients, and compared to healthy controls [89]. The authors analysed post-mortem cerebral cortex tissue from these patients and tried to match findings from transcriptomic analysis of lipid metabolic pathways with lipidomics screening. From the transcriptomic analysis, these authors identified alterations in peroxisome b-oxidation (higher levels of *ACAA1* and *ACOX3* mRNA in c9FTLD group compared to controls), bile acid biosynthesis (lower levels of *CYP27A1* mRNA in sALS and sFTLD-TDP, and much higher in c9FTLD, compared to controls), and biosynthesis of acylcarnitine (higher levels of *ACOT* mRNA in c9FTLD group compared to controls). Peroxisomes are an important site for cellular lipid metabolism, including β-oxidation of very-long fatty acids (VLCFA) and bile acid production. In the lipidomic analysis, there were higher levels of acylcarnitine, which links β-oxidation in the peroxisome and mitochondria, but the analysis failed to detect differential levels of VLCFA (22:0, 24:0), or bile acid. In general, not many differential lipid species were detected in the frontal cortex of these TDP43 proteinopathies; just 63 out of the total of 1119 lipid species identified were significantly different, and most of them showed decreased levels versus the controls; mainly, glycerophospholipids (phosphocholine PC; phosphoethanolamine PE) and sphingomyelin SM, which might reflect the neurodegenerative process in the brain. The main differences found among the different forms of TDP43 proteinopathies in the frontal cortex were in TG levels (mostly 18:1 oleic acid), which was higher in the ALS group compared to the FTLD-TDP group, and the increased levels of cholesterol ester CE (20:1) in the c9FTLD group. Interestingly, the two branched fatty acid esters of hydroxy fatty acids (FAHFAs) were decreased, which are related to glucose homeostasis and oxidative stress.

We summarize some remarkable communalities in the lipid alterations between ALS and FTD patients, mainly in some species of fatty acyls and glycerophospholipids: in blood samples, the levels of TGs, DGs, and VLCFAs are increased, while the levels of PS are decreased. In the neuronal tissue studies reported increased levels of TGs and MUFAs, and decreased levels of PC. However, some of these alterations could be a consequence of neurodegenerative process rather than a specific feature of the ALS–FTD spectrum. For example, the increase in TGs in the neuronal tissue found in ALS and FTD also has been described in AD [112]. Furthermore, in ALS there is an increase in the Cer levels in blood, as what happens in PD [113], as well as an increase in post-mortem brain tissue from AD patients [112,114]. 

### 3.4. Genetics

Genetic variants are found causally linked to FTD more often than in ALS, with up to 40% of FTD cases of familial, genetic origin and mostly of autosomal dominant inheritance. As in most complex neurodegenerative disorders, there is wide genetic heterogeneity between patients, with several genes and variants implicated, including mutations in the microtubule-associated protein tau gene (MAPT), the progranulin gene (GRN), and repeat expansion in the C9ORF72 gene; these three genes account for nearly 75% of all familial FTD cases. The pathological hexanucleotide repeat expansion in the C9ORF72 gene is the most common causal mutation in ALS, FTD, and mixed forms.

It is very intriguing to observe that patients carrying the exact same mutation, even within families, can develop ALS, bvFTD, or mixed forms. Thus, the genetic background of an individual, together with environmental risk factors, likely contribute to the manifestation of the different clinical forms observed, in both the genetic and sporadic forms.

GWAS studies and other association studies have been able to describe many modifiers or common variants with small effects that contribute to risk. In particular, regarding genes in relation to lipid metabolic regulation, the genetic variants for the APOE gene [115] have a role in determining clinical forms of FTD, with APOE2 being protective and APOE4 increasing the risk of FTD, although these associations have not been entirely reproducible [116].

There are not many mendelian association studies on FTD. We have recently conducted a Mendelian randomization association study on FTLD with TDP43 proteinopathy, the FTLD TDP subtype, to evaluate the potential causal association with genetically determined risk factors of lipid metabolism regarding body complexion and circulating lipids. We showed that, unlike what is found in ALS with LDL cholesterol association, no metabolic risk factors were strongly associated with FTD, rather just a suggestive association of higher level of triglycerides in blood with increased risk of the FTLD TDP subtype, which needs further research [98]. These suggestive findings are interesting since higher levels of TGs are found in many bvFTD patients.

## 4. Discussion

We gathered the main findings regarding systemic lipid metabolic alteration in ALS and in bvFTD, mostly from association studies conducted in the patients. In ALS, the disease progresses with weight loss and changes in fat depot distribution, with an increase in visceral fat. On the contrary, patients suffering bvFTD showed higher weight and increased adiposity, and also increased visceral fat depots. Those body changes are accompanied by alterations in the lipid composition in blood.

In the blood of ALS and FTD patients, many studies have found higher levels of total fatty acids, TGs, and sterol lipids, and decreases in sphingomyelins and phospholipids, although the particular lipid composition might not be the same among the different studies. These communalities in the lipids in the blood of ALS and bvFTD patients are remarkable, since the weight, food intake, and adiposity changes in these two ends of the spectrum of disorders are very different. It might be interesting to conduct a comparison with other neurodegenerative disorders to determine the general body responses to diseases. In the post-mortem frontal cortex of ALS and bvFTD TDP subtype patients, the most relevant common findings are the increased levels of cholesterols esters, oxysterols, ceramides, and sphingomyelins. At the same time, lipidomics might identify lipid species that would serve as biomarkers of the different ALS and FTD forms and clinical subtypes, helping in the stratification of patients.

Lipid alterations: cause or consequence? In some cases, observed lipid metabolic abnormalities seem to be a clear consequence of the disease process itself. In particular, weight loss and gain, as a consequence of altered food intake and diet preferences, as observed in ALS and FTD patients, respectively, together with hyper- or hypometabolic states naturally impact the systemic lipid metabolism of patients. At the cellular level, especially in neuronal tissues, cholesterol, sphingomyelins, phosphatidylcholine, and many other lipids are released from the membranes when cells die, which activates many of the inflammatory processes observed in these disorders. These inflammatory events can be found in most neurodegenerative disorders

A more causal or influential role for lipids in ALS and FTD disorders could arise from environmental interactions, diet, changes related to age, and are most likely influenced by genetics. The free fatty acids are only a minor fraction of the lipids in the blood and are mainly derived from adipose tissue and the composition is much related to the diet [117]. The release of fatty acids by adipose tissue is normally regulated depending on the energy demands of the body. In the cases of ALS and FTD, this regulation is greatly diminished, reflected by hypermetabolism, abnormal adipose tissue deposits, and alterations in the leptin and hypothalamus signalling pathways [39,40,41,99]. The fatty acid composition of the brain is then also adapted from the composition of the blood, thus linking systemic regulation and diet with brain composition. It is thus not surprising that diet might influence brain function through lipid metabolism regulation. In AD, increased risk of disease and cognitive decline have been associated with higher saturated fatty acids in the diet [118]. The TG levels in the blood depend on the diet and are distributed to tissues by lipoproteins. It is interesting to note that in the case of FTD, dietary changes and higher food intake might directly affect the TG levels and fatty acyls selection, promoting alterations in these lipids observed in patients.

We have previously attempted to hypothesize a potential causal role for lipid homeostasis dysregulation in neuronal tissues of ALS patients. In highly metabolic neurons, the production of reactive oxygen species (ROS) could accumulate with time, as a consequence of specific inherited genetic lesions, or stochastically as part of the ageing process. High ROS production would eventually affect the formation of oxysterols and other lipid peroxidation events that damage membranes, including the formation of inflammatory eicosanoids species, eventually disrupting the lipid homeostasis of these cells and tissues, potentially contributing to pathological mechanisms, leading to development of disease symptoms.

The genetic predisposition of some patients, by particular genetic variants directly linked to lipid dysregulation, is another entry point for the lipid metabolic abnormalities observed broadly in neurodegenerative disorders. There are several examples, including the genetic variant *APOE4* associated to AD. Mutations in *CYP27A1* cause a rare lipid storage disorder (cerebrotendinous xanthomotosis) with intellectual disabilities, dementia, ataxia, and epilepsy, among other systemic symptoms. It is important to highlight that several mendelian randomization studies have consistently found that the genetically determined higher levels of LDL in blood are associated with a higher risk of ALS [90,91,92]. It is worth exploring further the genetic–environment interaction that might be so important for ALS.

The different lipid classes are interconnected. The regulation of lipid metabolism is very complex, not only regarding systemic and neuronal tissue regulation, but also in health and disease situations. The attempts to find “the biomarker” or “the pathway” or “the molecule” for a complex disorder such as ALS might be challenging, since the regulation of lipid metabolism pathways are largely interconnected. Instead, the lipid profile seems a better way of reporting changes thanks to the more extended -omics analysis of lipids (lipidomics in combination with other -omics) in the given tissue analysed. Researchers might need to develop a systemic, holistic view of the changes observed in these disorders, for which we should introduce into our research more bioinformatic analysis tools that could assist in the study of these complex disorders.

The use of different tissues for the lipid analysis would naturally give distinctive types of information that could be complementary and help to provide a holistic view of the lipid metabolic regulation systemically. Saying this, each of the three tissues analysed that have been discussed here, bring some inherited limitations.

One of the most prevalent findings from past studies has been observed alterations in cholesterol, especially at the systemic level in blood samples measured by classical clinical biochemistry, in the form of lipoproteins (LDL, HDL, or TC). There is a very good review that summarizes the studies conducted on ALS patients from 2008 and the main findings on LDL, HDL and TC levels, highlighting the still remaining controversy in the levels of cholesterol in the blood and CSF of ALS patients [119]. What seems to be consistent among some studies that focussed on presymptomatic disease stages is that cholesterol levels in blood are elevated years before disease onset, which could be linked to the timing of disease appearance and pathology.

The variability and discrepancies in the analysis of blood lipids by classical clinical biochemistry is not only restricted to ALS and FTD but also can be found in the majority of the neurodegenerative disorders. In particular, the suggestion that elevated serum levels of TG, LDL, and TC are protective factors in PD patients has been in debate for a few years now [120,121], Therefore, the study of the predictive value of blood lipids for prognosis of neurodegenerative diseases is still going on. These controversies are not only due to the different analytical tools use by the different laboratories around the world, but also to the heterogeneous clinical population, sample collection, disease stage, etc.

At the neuronal tissue level, alterations in cholesterol homeostasis could be coming from different sources. Others and we have suggested that the homeostasis of cholesterol in the neuronal tissue seems to be locally altered from early disease stages. The higher cholesterol level and some oxysterol species (such as 25-HC) could be reflecting alterations in the endogenous production of cholesterol, since the higher cholesterol levels found in the CSF of ALS patients were not those imported from circulation, suggesting also a potential impairment in the efflux of cholesterol [59]. Cholesterol is an integral component of the membranes and myelin sheets. Cholesterol dyshomeostasis is found in many neurodegenerative disorders, from AD to PD and Huntington’s disease (HD) [10,122], amongst others, including ALS. In particular, elevated levels of the oxysterol 24-OHC have been found increased in many of these disorders, including PD, AD and HD [123,124]. These common alterations have led to suggest that the 24-OHC could be a marker of alteration of the metabolically active neurons. Thus, in this context, many of the brain cholesterol homeostasis alterations observed in many disorders seem to be secondary to the disease process, rather than a causative trigger. Still, the suggestive genetically determined association of cholesterol with ALS cannot be overlooked, as we previously showed that 89 SNPs associated with ALS are involved in cholesterol biosynthesis by pathway analysis [64]. Thus, proper management of the cholesterol pathway alterations could be a target that deserves further consideration in order to modify disease progression.

The hyperlipidaemia observed in some ALS patients could promote a better prognosis of the disease according to some studies [48,52]. Some of these studies using high-calorie diets have had promising results in patients and in animal models, but it remains to be determined how much and what type of calories are exerting the possible benefit Therefore, there are still several clinical trials based on high-fat diets on ALS (Clinicaltrials.gov ID: NCT00983983, NCT02306590, NCT04172792). Studies that have an opposite approach, such as lowering the cholesterol levels by statins, have not given good results so far, reinforcing the idea that high levels of cholesterol might be a compensatory mechanism in ALS patients [69,70]. However, not all lipid increases are beneficial. One study has detected up to a 2.5-fold increase in arachidonic acid in ALS patients [82]. This lipid is the precursor of eicosanoids (thromboxane, leukotrienes, and prostaglandins), which are lipids that regulate many inflammatory processes, whose deregulation could lead the pathological mechanism of neuroinflammation.

Sphingolipids are highly deregulated in ALS patients. The central metabolites of this lipid group are ceramides, and these are significantly elevated in the three tissues analysed in here. Ceramides are actively involved in cell signalling and high levels trigger apoptosis. The accumulation of ceramides observed in the spinal cord of ALS patients is thought to contribute to the death of motor neurons. There is currently a clinical trial in ALS patients with a compound called Fingomilod, which is a sphingosine 1-phosphate receptor antagonist (Clinicaltrials.gov ID: NCT01786174). Blocking these receptors prevents the infiltration of T lymphocytes into the CNS and thus reduces neuroinflammation.

With all these data, a promising therapeutic avenue for ALS patients emerges that uses lipid metabolism dysregulations as a target among which are (a) restoring homeostasis of cholesterol metabolism in the CNS; (b) promoting the beneficial effects of blood hyperlipidaemia; (c) decreasing the arachidonic acid levels to decrease the synthesis of proinflammatory derivatives; and (d) decreasing the ceramide levels to inhibit pro-apoptotic signalling.

Lipidomics analysis of the different samples in several ALS studies are showing inconsistent results, as expected for the heterogeneity of disease forms and cohorts, and the analysis performed. The great advances in lipidomics is beset with some problems that hopefully will be soon resolved. However, for now, there are many discrepancies in the standardization of the methods. To start with, not all the equipment and mass spectrometers can analyse all type of lipids. For example, for fatty acids we need to use GS/MS, whereas other lipids might need LC/MS, and oxysterols derivates need a different method. Thus, the analysis of the whole lipidome is not possible now with one type of technique. In addition, we need to consider that some lipid species are at such a low level that their consistent and correct detection might not even be possible. There are, at the moment, three main methods for mass spectrometry analysis of the lipidome, two unbiased and one targeted: (1) shotgun lipidomics, which detects the larger number of lipids of the major species (newer equipment with higher resolution TOF/orbitrap are available); (2) chromatography mass spectrometry (LC-MS and LC-MS/MS), which has better separation and identification of species; and (3) targeted LC-MS/MS, with improved the identification and facilitation of the quantification analysis. The choice of references and internal lipid standards are also critical and would dramatically affect correct species identification and results. The subsequent analysis, statistical tools, determination, and quantification are again a great source of variation among labs. Finally, the biological and chemical interpretation of the data can also be a source for variation. There are increasing efforts to standardize all these methods and analytical tools that should help to reduce the high variation obtained with the current methods, e.g., the LIPID MAPS consortium (the LIPID MAPS^®^ Lipidomics Gateway, Available online: https://www.lipidmaps.org/, (accessed on 5 May 2022)), and many others are working extensively on these problems, and hopefully those would be overcome soon, which will help to obtain results that are more consistent and robust.

Perhaps surprisingly, there are fewer altered lipid species in the CSF compared to blood in ALS patients, versus age–sex matching healthy controls. However, there are typically fewer lipid species identified in CSF (200–800) versus blood (500–1000) in such studies. This could support the concept that ALS is not only a neurodegenerative disease, but rather a wider systemic disorder with many other organs affected, as reflected in the blood changes. The combination of blood and CSF lipidomics, conducted in parallel in the same individuals, could provide information on the potential crosstalk between systemic lipid metabolism and the CNS, especially since degeneration of the blood–brain barrier (BBB) in ALS patients might allow the traffic of certain metabolites and lipids that are restricted in healthy conditions. Not many differential lipid species have been found in common in the CSF and blood of ALS patients versus the controls, which might be related to heterogeneous disruption to the BBB found in ALS, found in around 18% of patients, the majority of which displayed a lumbar onset type [125]; this again highlights the need for better stratification in studies, where possible. In relation to neuronal tissue findings, the two major limitations here are the use of post-mortem tissue as well as the limitation of not being able to run single cell lipidomics, since that technology is still not available.

The study of lipid metabolism alterations as risk factors for ALS and FTD is more complex and needs extensive population-based studies, including genomic analysis. Previous population studies have classically used the weight and BMI data of the patient years before the onset of the symptoms. It is arguable that the weight data used from a few years before the debut might already be affected by the disorder, and thus it is not a good risk factor indicator but rather an early marker of the disease. The integration of the genome in these complex paradigms is helping to unravel the role of lipid metabolism in the risk of these disorders. Many more studies are needed to understand the impact of lifelong lipid metabolism alterations as risk factors for ALS and FTD, combining all the data from the clinical, metabolic, habits, environmental, and genetics factors in these complex disorders.

## 5. Conclusions

Lipid metabolism regulation across the lifetime of an individual is an important factor that seems to impact differentially on the risk of suffering from ALS or FTD or the mixed forms. Here, we propose the inclusion of more systematic metabolic tests in routine neurological clinics, which would help to draw more solid conclusion in future studies. To date, most studies have focussed on one or two parameters in their studies, but a more comprehensive integration of data (BMI, adiposity and body composition, lipidomics, nutritional status, energy consumption, and genetics, among others) should be used to provide a better view of the relation between metabolism and these disorders. Similarly, different lipidomic studies evidenced the complex lipid regulation, not only systemically but also in the neuronal tissue, as well as the inconsistencies in finding specific lipid biomarkers for diseases, which should make us think that future research might need to use several integrative tools to provide a more systemic holistic view of the disorders.

From these systemic approaches, we then would be able to disentangle partly the impact of the external regulation of lipid metabolism on the spinal cord and brain in healthy and diseased individuals, which could drive new intervention strategies to treat ALS and FTD, as well as to work on prevention.

## Figures and Tables

**Figure 1 biomedicines-10-01105-f001:**
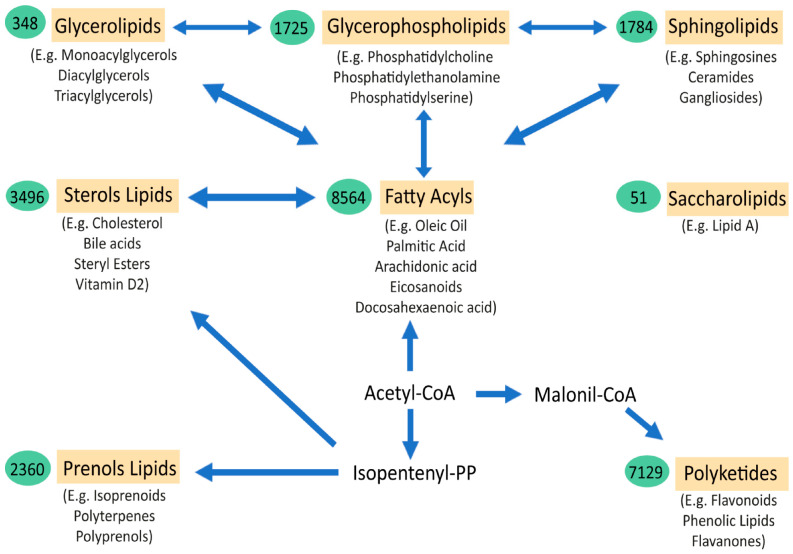
Schematic representation of the major lipid classes and their relations. The relationships between the main categories of mammalian lipids, starting with the 2-carbon precursor acetyl CoA, which is the basis for fatty acid biosynthesis, have been observed. Fatty acyls, in turn, through various modifications, give rise to complex lipids, such as sphingolipids, glycerolipids, glycerophospholipids, and sterols. Another route to generate other classes of lipids from acetyl-CoA is through isopentenyl pyrophosphate, which provides the building blocks for prenols and sterol lipids. Finally, the biosynthesis from acetyl-CoA, via conversion to malonyl-CoA, gives rise to polyketides. Saccharolipids are an unrelated group of lipids that are found in bacteria. The arrows denote multi-step transformations between the main lipid categories from acetyl CoA, isopentenyl pyrophosphate, and malonyl-CoA. The values in the green ovals represent the number of lipid structures curated within each lipid category. Below each lipid class are examples of the most representative molecules of the group (Adapted from Quehenberger et al. JLR 2010 [2]).

**Figure 2 biomedicines-10-01105-f002:**
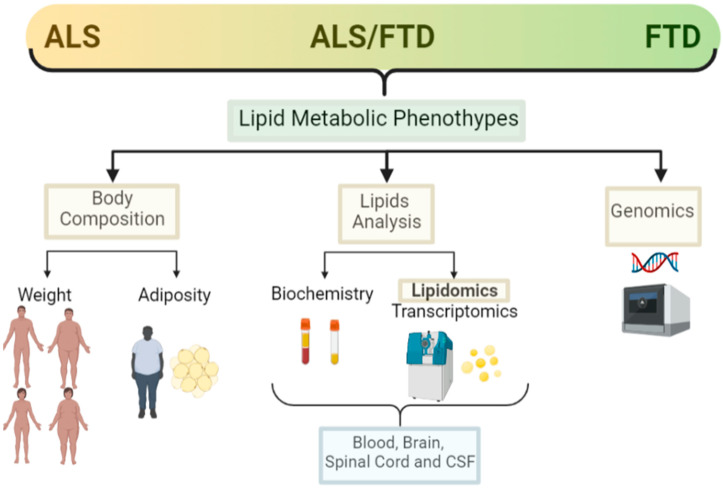
Lipid metabolic clinical features in the ALS–FTD spectrum of disorders. Schematic representation of the main clinical features analysed in ALS and FTD patients in relation to lipid metabolism and the techniques used for those analysis (created with Biorender).

**Figure 3 biomedicines-10-01105-f003:**
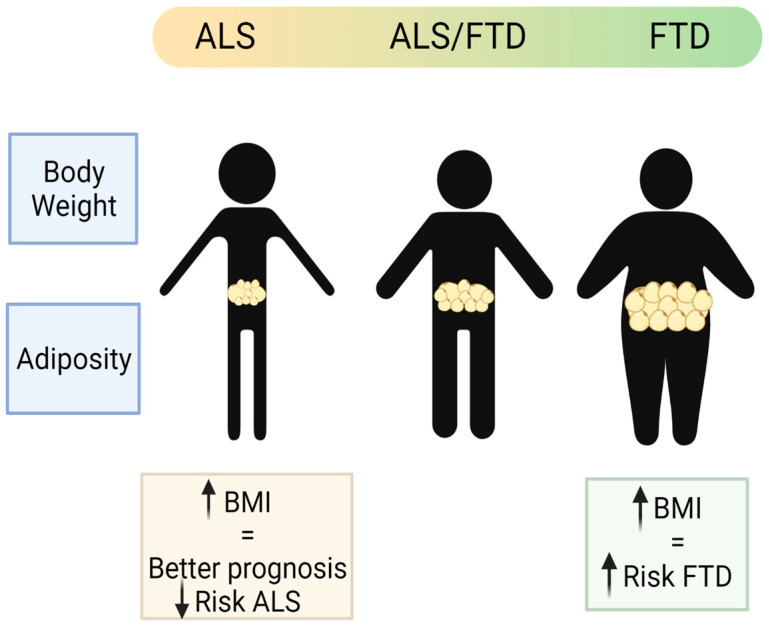
Schematic representation of the main body weight and adiposity features in the patients of the ALS–FTD spectrum of disorders (created with Biorender).

**Table 1 biomedicines-10-01105-t001:** Summary of the main findings in the lipid analysis in the blood, CSF, and neural tissue from ALS and FTD patients.

Lipid Class	Tissue	ALS	FTD
Sterols Lipids	Blood	Up: LDL, LDL/HDL, 25-OHC Down: 27-OHC Refs. [52,53,54,65,67]	Up: Down: HDL Refs. [90,91]
	CSF	Up: 25-OHC Down: TC, 24-OHC Refs. [59,65]	n.d.
	Neuronal Tissue	Up: CEs, 24-OHC, 7α-OHC Down: Refs. [60,61,66]	Not significant changes found Ref. [89]
Fatty Acyls	Blood	Up: TGs, DGs, VLCFA Down: MGs Refs. [53,56,74,75,77,78]	Up: TGs, DGs, VLCFAs Down: Refs. [91,92,93,94,95]
	CSF	Up: Down: TGs Refs. [79,88]	n.d.
	Neuronal Tissue	Up: MUFAs, TGs Down: PUFAs, SFAs Refs. [61,81]	Up: TGs, MUFAs, PUFAs Down: Refs. [92,94]
Sphingolipids	Blood	Up: Cer, GlycoSph Down: SMs Refs. [56,78,79]	Not significant changes found Ref. [93]
	CSF	Up: GlycoSph, SMs Down: Refs. [79,86,88]	n.d.
	Neuronal Tissue	Up: Cer, GlycoSph, SMs Down: Refs. [60,85]	Up: Down: SMs Ref. [89]
Glycerophospholipids	Blood	Up: PE Down: PC, PCe, PS, Pep Refs. [56,78]	Up: Down: PS, PG Refs. [93,94]
	CSF	Up: PC Down: Refs. [79,88]	n.d.
	Neuronal Tissue	Up: Down: PC Ref. [89]	Up: Down: PC, PE Ref. [89]

TC: Total Cholesterol; LDL: Low Density Lipoprotein; HDL: High Density Lipoprotein; CEs: Cholesterol esters; TGs: Triglycerides; DGs: Diglycerides; MGs: Monoglycerides; VLCFAs: Very-long-chain fatty acids; Cer: Ceramides; Glycosph: Glycosphingolipids; PUFA: Poly-Unsaturated Fatty Acids; MUFA: Mono-Unsaturated Fatty Acids; SFa: Saturated Fatty Acids; SMs: Sphingomyelins; PC: Phosphatidylcholine; PS: Phosphatidylserine; PE: Phosphatidylethanolamine; PG: Phosphatidylglycerol; PCe: Phosphatidylcholine ethers; PEp: Phosphatidylethanolamine plasmalogen; 7α-HC: 7-alpha-Hydroxy-Cholesterol; 24-HC: 24-Hydroxy-Cholesterol; 25-HC: 25-Hydroxy-Cholesterol; 26-HC: 26-Hydroxy-Cholesterol; 27-HC: 27-Hydroxy-Cholesterol; n.d.= Not determined. The numbers in bracket are the references of the studies where those results are reported.

## Data Availability

Not applicable.

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
