# Peer review of "Lipid Metabolic Alterations in the ALS–FTD Spectrum of Disorders"

_biomedicines, 2022, doi:10.3390/biomedicines10051105_

Round 1

Reviewer 1 Report

This is a review article about lipid abnormalities in ALS, FTD and ALS/FTD complex. The review as presented is an encyclopedic examination of published work about these severe and disabling, fortunately rare, conditions.

No attempts were made to discuss how these reported abnormalities could arise biochemically. Perhaps in the Introduction, the authors could include a Figure that educates the reader about metabolism of the various lipid species and how their metabolisms interconnect (or not).

Unfortunately, the authors report no consistent trends, but they also have not identified preponderant abnormalities. They also did not examine other more common neurodegenerative conditions, although towards the end of their paper they note how comparison to these other disorders might be helpful. I searched PubMed for numbers of references (realizing that not all references may be exactly "on topic"), using the search term "lipid metabolism in X disease" and found that between 2000 and 2022 PubMed listed 3970 articles for Alzheimer's disease, 1632 articles for Parkinson's disease, 292 articles for ALS and 25 articles for FTD. Surely there must be enough data for the authors to compare and contrast lipid metabolism in the more commonly occurring neurodegenerations (AD and PD) compared to the rare conditions ALS and FTD with or without ALS.

I also found interpretation of Table 1 to be difficult. What is increased or decreased in each category of lipid is not clear.

For the above reasons I find the review not helpful. I propose that the authors undertake major revisions to address the above deficits. This area of lipid metabolism in neurodegeneration is a very important topic that is not discussed often enough, and the authors need to present a review worthy of that deficiency.

Reviewer 2 Report

In the the manuscript "Lipid metabolic alterations in the ALS-FTD spectrum of disorders" by Juan Miguel Godoy-Corchuelo et al. the authors attempted to summarize new relevant information about the lipidomics in ALS-FTD spectrum of disorders. The theme is interesting and extremely relevant for the field. In addition, the paper is well-written and organized. However, the paper could benefit from a few improvements:

  1. When describing some important topics, the authors fail to cite relevant literature. The authors should thoroughly review the paper and add relevant literature where needed. For example, that is the case when addressing the clinical aspects of ALS: “Amyotrophic lateral sclerosis is a fatal, rapid, progressive neurodegenerative disorder characterised by loss of upper and lower motor neurons of the frontal cortex and spinal cord, leading to a loss of capacity to move, talk (dysarthria), swallow (dysphagia) and ultimately breath (due to weakness of the respiratory muscles), requiring tube feeding and mechanical ventilation at late stages”. The authors could cite here comprehensive previous review papers like:

- Brown, R. H., and Al-Chalabi, A. (2017). Amyotrophic Lateral Sclerosis. N. Engl. J. Med. 377 (2), 162–172. doi:10.1056/NEJMra1603471

- Lamas, N.J.; Roybon, L. Harnessing the potential of human pluripotent stem cell-derived motor neurons for drug discovery in Amyotrophic Lateral Sclerosis (ALS): From the clinic to the laboratory and back to the patient. Front. Drug Discov. 2021, 1, 1–26.

- Kiernan, M. C., Vucic, S., Talbot, K., McDermott, C. J., Hardiman, O., Shefner, J. M., et al. (2020). Improving Clinical Trial Outcomes in Amyotrophic Lateral Sclerosis. Nat. Rev. Neurol. 17, 104–118. doi:10.1038/s41582-020-00434-z.

  1. The role of statins in ALS is addressed in one part of the text in a lipid-centered perspective [“Due to the alterations in blood cholesterol levels observed in ALS patients, several groups have evaluated the treatment efficacy of cholesterol-lowering statins in the risk of ALS. There are some controversies in these results, with mainly no effect, or a negative effect, on the progression of the disease (60,61). Similarly, in mice, the use of statins to lower cholesterol levels accelerates disease progression and decreases survival in the SOD1G93A transgenic models (62)”]. However, statins have shown other effects in motor neurons besides changes in lipidic profile. The authors should enrich the discussion and cite key literature such as:

- Li, H., Kuwajima, T., Oakley, D., Nikulina, E., Hou, J., Yang, W. S., et al. (2016). Protein Prenylation Constitutes an Endogenous Brake on Axonal Growth. Cel Rep. 16 (2), 545–558. doi:10.1016/j.celrep.2016.06.013.

  1. The challenges in lipidomics analysis are only lightly discussed. The authors should expand the discussion about this topic, addressing key issues regarding patient samples, different methodologies of lipidomic analysis, etc... That will certainly enrich the discussion about this topic.

  1. The paper could also benefit if some information about possible/promising drug targets aimed at lipidic modulation in ALS are discussed.

  1. Finally, the current design of table 1 is confusing: the way the arrows are placed does not allow to easily interpret the changes that are being summarized. Please build a table that is more clear for the reader to digest all the information.

Therefore, the present review paper summarizes relevant infomation about lipidomics in ALS/FTD, but it will be mandatory to perform additional changes in the present manuscript if it is considered relevant for publication in Biomedicines.

Reviewer 3 Report

The current review focused on the Lipid metabolic alterations in the ALS-FTD spectrum of disorders.  They mentioned that ALS presentation is within a disorder spectrum with Frontotemporal dementia (FTD), and many patients present mixed forms of ALS and FTD. It is very well written and included comprehensive information. They first discussed about the role of fatty acids in ALS. They also discussed the role of fatty acids in other neurodegenerative disorders. Then they aimed to understand several factors such as BMI, Adiposity. They also categorized lipids and related metabolites and receptors to relate to the pathophysiology of ALS.

Author Response

Thank you very much for you revision and kind words.

Round 2

Reviewer 1 Report

This manuscript has been extensively revised, mainly by addition of many detailed paragraphs. I feel it is now a major reference for the field (mostly ALS, as the authors admit to in several places) and has the potential to stimulate critical thinking about therapeutic strategies towards using lipidomics for ALS. The last section that points out the limitations of various analytical methods is very important, and at some later point, the authors may wish to write a "position paper" suggesting a core battery of lipidomic tests (and best analytical approaches) for individuals with ALS. Perhaps there are subgroups with specific abnormalities?

Author Response

Thank you very much for your comments and suggestions. 

We will most likely continue reporting the most interesting tools and developments requiered for each lipid specific analysis, based on our experience and others,  as we are currently conducting different lipidomics analysis in our research and we are facing and trying to solve all these issues.  

As the reviewer is suggesting, we also believe that there are subgroups of patients for the lipid regulation abnormalities observed that will be interesting to explore in the context of personalized medicine in future and more focussed studies.